# Involvement of Lysophospholipids in Pulmonary Vascular Functions and Diseases

**DOI:** 10.3390/biomedicines12010124

**Published:** 2024-01-08

**Authors:** Hiroaki Kume, Rina Harigane, Mami Rikimaru

**Affiliations:** Department of Infectious Diseases and Respiratory Medicine, Fukushima Medical University Aizu Medical Center, 21-2 Maeda, Tanisawa, Kawahigashi, Aizuwakamatsu City 969-3492, Fukushima, Japan; lu7jdq@fmu.ac.jp (R.H.); mami-r@fmu.ac.jp (M.R.)

**Keywords:** lysophosphatidic acid, lysophosphadidylcholine, sphingosine 1-phosphate, pulmonary vascular smooth muscle, pulmonary endothelial cells, pulmonary fibroblasts, pulmonary fibrosis, pulmonary hypertension, ARDS, asthma

## Abstract

Extracellular lysophospholipids (lysophosphatidic acid, lysophosphatidylcholine, sphingosine 1-phosphate, etc.), which are synthesized from phospholipids in the cell membrane, act as lipid mediators, and mediate various cellular responses in constituent cells in the respiratory system, such as contraction, proliferation, migration, and cytoskeletal organization. In addition to these effects, the expression of the adhesion molecules is enhanced by these extracellular lysophospholipids in pulmonary endothelial cells. These effects are exerted via specific G protein-coupled receptors. Rho, Ras, and phospholipase C (PLC) have been proven to be their signaling pathways, related to Ca^2+^ signaling due to Ca^2+^ dynamics and Ca^2+^ sensitization. Therefore, lysophospholipids probably induce pulmonary vascular remodeling through phenotype changes in smooth muscle cells, endothelial cells, and fibroblasts, likely resulting in acute respiratory distress syndrome due to vascular leak, pulmonary hypertension, and pulmonary fibrosis. Moreover, lysophospholipids induce the recruitment of inflammatory cells to the lungs via the enhancement of adhesion molecules in endothelial cells, potentially leading to the development of asthma. These results demonstrate that lysophospholipids may be novel therapeutic targets not only for injury, fibrosis, and hypertension in the lung, but also for asthma. In this review, we discuss the mechanisms of the effects of lysophospholipids on the respiratory system, and the possibility of precision medicine targeting lysophospholipids as treatable traits of these diseases.

## 1. Introduction

Lysophospholipids, which are phospholipids that have one acyl group, are classified into two types, i.e., those with a glycerol backbone and those with a sphingosine backbone. Lysophospholipids have simple structures composed of a single hydrophobic fatty acid chain, a hydrophilic head, and a phosphate group with or without a large molecule attached. Lysophospholipids are categorized based on their head group and fatty acid chain, i.e., lysophosphatidic acid (LPA), lysophosphatidylcholine (LPC), lysophosphatidylserine (LPS), and sphingosine-1-phosphate (S1P) (Figure 1), and each compound has characteristic physiological activities. Lysophospholipids are synthesized from phospholipids in the cell membrane and are released into the extracellular environment because of their relative hydrophilicity. Previous studies have identified G protein-coupled receptors (GPCRs) that respond specifically to certain lysophospholipids in many types of organs. Extracellular lysophospholipids act as lipid mediators, and mediate various cellular responses, such as proliferation, migration, apoptosis, and cytoskeletal organization. In addition to these effects, the expression of adhesion molecules is augmented in pulmonary artery endothelial cells. Vascular leak is enhanced via barrier dysfunction through cytoskeletal reorganization in endothelial cells, and the recruitment of inflammatory cells to the lungs is facilitated by adhesion molecules in endothelial cells. Moreover, lysophospholipids have effects on Ca^2+^ signaling (Ca^2+^ dynamics and Ca^2+^ sensitization), resulting in alterations in the contractility of smooth cells. Therefore, lysophospholipids may play important roles in the pathophysiology relating to inflammatory diseases in the respiratory system such as acute respiratory distress syndrome (ARDS), pulmonary fibrosis, pulmonary hypertension, and asthma [1]. Lysophospholipids may contribute to these disorders through the regulation of oxidative stress [2,3,4]. Furthermore, recent reports have indicated the possible involvement of lysophospholipids in inflammatory processes related to the pathophysiology of COVID-19 [5,6,7]. Therefore, lysophospholipids probably serve as therapeutic targets and pathological biomarkers for various diseases [8].

Clinical reports have indicated that the concentrations of LPA and S1P in bronchoalveolar lavage fluids (BALFs) are markedly augmented in patients with asthma after allergen challenge [9,10]. LPA and S1P also contract the airway and vascular smooth muscle through Ca^2+^ dynamics [11,12], and they increase the response to a muscarinic receptor agonist in airway smooth muscle through Rho-induced Ca^2+^ sensitization [12]. Moreover, LPA and S1P enhance adhesion molecules in pulmonary vascular endothelial cells [13,14]. LPA levels are increased in BALFs following lung injury in a bleomycin model of pulmonary fibrosis [15]. The inhalation of LPA and LPC results in the recruitment of eosinophils to the lungs in guinea pigs [16,17], and LPC does not generate tension but causes hyporesponsiveness to β_2_-adrenergic actions without a change in the intracellular Ca^2+^ concentration in airway smooth muscle [18]. Lysophosphatidylserine may induce a type I (IgE-dependent) allergic response [19], but its involvement in pulmonary vessels is unknown. Lysophospholipids, such as LPA, LPC, and S1P, are probably involved in the pathophysiology of the various inflammatory respiratory diseases shown above, and they have the potential to be novel therapeutic targets for these diseases, although the clinical relevance of lysophospholipids still remains unclear. Therefore, the advancement of therapy for these incurable respiratory diseases related to pulmonary vessels probably requires precision medicine to be established based on treatable traits such as lysophospholipids.

This review states the physiological activities of lysophospholipids in the constituent cells of the pulmonary vessels, such as smooth muscle cells, endothelial cells, and fibroblasts. Moreover, the involvement of lysophospholipids in the pathophysiology of these incurable respiratory diseases is also discussed.

## 2. Lysophospholipids

### 2.1. Structure

Lysophospholipids are synthesized via phospholipase A1, A2, C, and D (PLA1, PLA2, PLC, and PLD) from phospholipids, such as phosphatidic acid (PA) and phosphatidylcholine (lecithin), and they have a hydrophilic “head” containing a phosphate group and a hydrophobic “tail” derived from long fatty acid residues, different from phospholipids due to a lack of a fatty acid chain in either sn-1 or sn-2. Lysophosphatidic acid (LPA) is the simplest lysophospholipid, consisting of a glycerol backbone with the addition of a phosphate group at the sn-3 position and a hydroxyl group and a fatty acid chain in either the sn-1 or sn-2 position, with their variations being due to different fatty acid chains (Figure 1). The phosphate group can be modified with simple organic molecules, i.e., lysophosphatidylcholine (LPC) depends on choline attached to the phosphate group (Figure 1). Sphingosine-1 phosphate (S1P), which has various physiological activities, is another class of lysophospholipids, and it is synthesized from sphingosine through phosphorylation by sphingosine kinase (SphK) in the process of metabolism of sphingolipids, another class of phospholipids. Sphingolipids are characterized by containing a backbone of sphingoid bases, which are long-chain aliphatic amines with two or three hydroxyl groups. Sphingosine, which is synthesized from ceramides by ceramidase, is a typical and the simplest sphingoid base, and ceramides, which also have physiological activities, contain sphingoid bases linked to fatty acids by amide bonds. S1P has a structure in which a phosphate group is bonded to sphingosine (a sphingoid base) (Figure 1).

### 2.2. Physiological Activity

Extracellular lysophospholipids exert wide range of physiological activities in various cell types via specific receptors, which are seven-transmembrane GPCRs [20]. LPA and S1P, as high affinity ligands, are connected to characterize receptor, that are LPA and S1P receptors, respectively. LPA receptors are currently classified into six different classes (LPA_1–6_), and LPA_1–3_ belong to the endothelial differential gene (Edg_2,4,7_) type, respectively. In contrast, LPA_4–6_ are non-Edg types. S1P receptors are also currently classified into five different types (S1P_1–5_), and S1P_1–5_ belong to the Edg_1,5,3,6,8_ type, respectively. Specific receptors for LPC, such as G2A and GPR4, are different types of GPCRs from LPA and S1P [21,22] (Figure 2). Moreover, peroxisome proliferator-activated receptor gamma (PPARγ) is identified as a receptor for intracellular LPA [23].
Figure 2The receptors and intracellular signal transduction processes of lysophospholipids. LPA: lysophosphatidic acid; LPC: lysophosphatidylcholine; S1P: sphingosine 1-phosphate; SOCE: store-operated calcium entry; TRPC: transient receptor potential channels; PLC: phospholipase C; IP_3_: inositol 1,4,5-trisphosphate; PKA: protein kinase A; ERK: extracellular signal-regulated kinase; PI3K: phosphoinositol 3-kinase; Akt: protein kinase B. Illustrated based on Refs. [20,21,22,24,25].
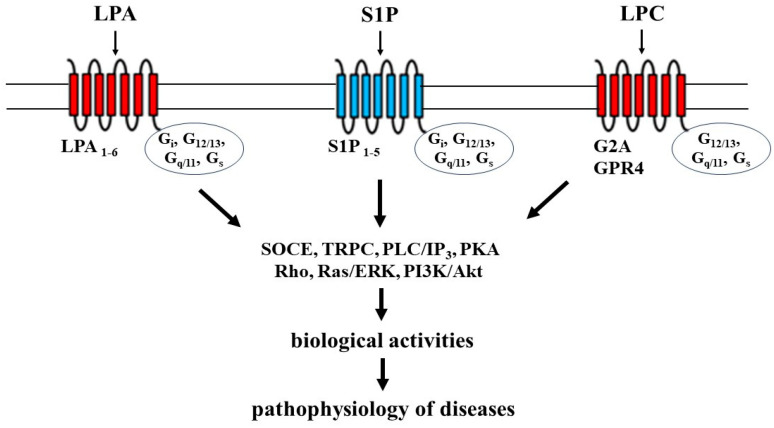



The α subunit of G proteins (G_α_) plays a major role in determining downstream signaling. Four distinct classes of the α subunit of G proteins (G_12/13_, G_q/11_, G_i_, and G_s_) are connected to the LPA and S1P receptors, leading to the downstream effector pathways in cells as described below [24,25] (Figure 2). G_12/13_ activates Rho (a small monomeric G protein)/Rho-kinase (a target protein of Rho) processes, resulting in cytoskeletal reorganization and Ca^2+^ sensitization. G_q/11_ activates phospholipase C (PLC)/inositol trisphosphate (IP_3_) processes, resulting in Ca^2+^ dynamics. G_i_ activates Ras/extracellular signal-regulated kinase (ERK) processes and phosphoinositol 3-kinase (PI3K)/Akt processes. G_s_ activates adenylyl cyclase/protein kinase A (PKA), resulting in phosphorylation. These intracellular signal transduction pathways lead to physiological and pathophysiological effects, such as contraction, migration, proliferation, survival, cell–cell interaction, and cytoskeletal reorganization in cells of various tissues. The mechanisms of receptor/G protein complexes and the conformation of receptors related to ligand recognition were unclear in terms of the physiological activities of external lysophospholipids until quite recently. However, these mechanisms have been elucidated using an analysis of the cryo-electron microscopy structure mainly regarding S1P_1_ and LPA_1_ [26,27].

## 3. Lysophosphatidic Acid

### 3.1. Structure and Function

Lysophosphatidic acid is the simplest bioactive phospholipid that is synthesized in the cell membranes of various tissues including in endothelial cell, and it exerts potent extracellular signaling through its interaction with its six specific GPCRs as a lipid mediator, resulting in cell proliferation, migration, contraction, and cytoskeletal reorganization. LPA consists of a glycerol backbone with the addition of a phosphate group at the sn-3 position and a hydroxyl group and a fatty acid chain in either the sn-1 or sn-2 position, with variations being due to different fatty acid chains. The most common LPA form known in research is described as 18:1 (with an 18-carbon chain length and 1 unsaturated bond). LPA receptors (LPA_6_) recognize ligands (LPA) through conformational rearrangement (inward shift of transmembrane helices 6 and 7) [28]. Extracellular LPA is produced from lysophospholipids, such as LPC and LPS, by the enzymatic activity of extracellular autotoxin (ATX), which has lysophospholipase D activity and is classified as ectonucleotide pyrophosphatase-phosphodiesterase 2 (ENPP2), and it is also produced from PA by PA-selective PLA_1_. ATX is derived from the epithelium and macrophages. In addition to this process, LPA is generated from PA by PLA1/PLA2 independent of ATX. ATX is generally considered to be a major source of generated LPA, and extracellular LPA plays an essential role in physiological activities as a lipid mediator via cellular responses through LPA receptors. LPA receptors are widely expressed in several organs, including the brain and lungs. The activation of LPA_1,2_, which is connected to Rho/Rho-kinase processes, causes contraction in muscle cells via an increase in the response to intracellular Ca^2+^ (Ca^2+^-independent contraction), referred to as Ca^2+^ sensitization [29,30]. Since Rho/Rho-kinase processes also cause the dynamic reorganization of the actin cytoskeleton in non-muscle cells, the activation of LPA_1,2_ is associated with alterations in cell motility and morphology, resulting in cell migration and the dysfunction of cell–cell adhesion. Therefore, LPA causes phenotype changes in constituent cells of the respiratory system, i.e., smooth muscle contractility, fibroblast and inflammatory cell recruitment, epithelial apoptosis, and endothelial permeability [24]. A recent report has indicated that LPA_5_ may be a novel target for the development of agents for inflammatory and malignant diseases [31].

### 3.2. Effects on Smooth Muscle Cells

Since the Rho/Rho-kinase pathway is the intracellular signal transduction system of LPA receptors, except for LPA_3_, smooth muscle in the respiratory system may generate tension via Rho-mediated Ca^2+^ sensitization in the presence of LPA (Ca^2+^-independent contraction). A previous report has demonstrated that LPA enhances the contractility of isolated tracheal smooth muscle in rabbits and cats [32]. However, little is currently known about the contractile effects of LPA on pulmonary smooth muscle in detail. LPA generates tension in endothelium-denuded aorta segments [33]; moreover, LPA prevents endotoxin-induced dilatations in the pulmonary artery via the disassembly of the smooth muscle cell (SMC) F-actin cytoskeleton via the activation of the Rho/Rho-kinase pathway [34] (Figure 3), suggesting that LPA may cause contraction in pulmonary vascular smooth muscle, similar to airway smooth muscle [32]. Since the PLC/IP_3_ pathway is the intracellular signal transduction system of LPA receptors, except for LPA_6_, LPA may cause the contraction of pulmonary vascular smooth muscle via Ca^2+^ dynamics (Ca^2+^-dependent contraction). LPA increases the intracellular Ca^2+^ concentration in cultured A10 vascular smooth muscle cells through store-operated Ca^2+^ entry (SOCE) via the PLC/IP_3_ pathway, a Na^+^-Ca^2+^ exchanger, and a Na^+^-H^+^ exchanger [11] (Figure 3). However, it is debatable whether LPA’s directly induced contraction in pulmonary vascular smooth muscle is involved in the pathophysiology of pulmonary hypertension. LPA enhances proliferation and migration in the vascular smooth muscle cells of the aorta and carotid artery [35,36,37] (Figure 3). Blood vessel restenosis is associated with proliferation and migration in vascular smooth muscle cells, which may be mediated by LPA during restenosis. When proliferation and migration are investigated using an electrical cell–substrate impedance sensor (ECIS), an inhibitor of ATX (PF8380) and an inhibitor of LPA receptors (Ki16425) attenuate proliferation and migration in carotid arterial smooth muscle cells in mice [37]. Cell proliferation and cell migration in smooth muscle cells may cause pulmonary arterial remodeling, resulting in pulmonary hypertension.

### 3.3. Effects on Fibroblasts

LPA causes cytoskeletal reorganization and the proliferation of lung fibroblasts through GPCRs-mediated signal pathways, such as Rho, Ras, and PI3K, resulting in pulmonary fibrosis and pulmonary hypertension [24] (Figure 3). Both the serum levels of LPA and the perivascular expression of LPA are increased in a rat model of hypoxic pulmonary hypertension, which is characterized by structural changes in the vascular wall of pulmonary arteries [14]. Moreover, the expression of ATX is increased in coexistence with mast cell tryptase in these hypoxic rat lungs, indicating that mast cells are involved in the generation of LPA. Augmented ATX staining is observed in the lungs of bleomycin-treated mice, and ATX levels are also elevated in corresponding BALFs [38]. Serum from low-oxygen-challenged rats has higher chemoattractant effects on primary lung fibroblasts, indicating that LPA facilitates fibroblast migration [14]. Since this phenomenon is suppressed in the presence of inhibitors of LPA_1,3_, LPA contributes to hypoxia-induced remodeling in the pulmonary vasculature, resulting in pulmonary hypertension [14]. In LPA_1_-lacking mice, the development of pulmonary fibrosis is markedly prevented, and the absence of LPA_1_ causes a decrease in fibroblast recruitment, indicating that LPA_1_ causes pulmonary fibrosis via fibroblast recruitment [15]. This phenomenon may be excessive when injury leads to fibrosis rather than to repair.

### 3.4. Effects on Endothelial Cells

An aberrant vascular system is a characteristic feature of ATX knockout mice, indicating that LPA may be involved in the malformation of vasculature during development [39]. LPA is associated with the function of endothelial cells via the expression of angiogenesis and the modulation of their permeability [40]. These endothelial cell functions are impaired by LPA-induced barrier dysfunction, leading to cancer and various inflammatory diseases [41] (Figure 3). The LPA-induced barrier function in the endothelium is caused by Rho-mediated cytoskeletal reorganization due to stress fibers which consist of actin filaments. LPA_1_ expressed in endothelial cells is probably associated with increased vascular permeability in lung injury [15] (Figure 3). Endothelial cell migration, which is an essential component of angiogenesis and wound repair, is activated by LPA. Since this phenomenon is inhibited by pretreatment with pertussis toxin (an agent for the uncoupling of G_i_ from receptors) and LY294002 (an inhibitor of PI3K), endothelial cell migration induced by LPA is associated with the G_i_/PI3K pathway, which causes actin filament remodeling (cytoskeletal reorganization) through Akt [42,43] (Figure 3).

LPA also markedly augments the adhesive properties of endothelial cells in the pulmonary artery and mononuclear cells in the peripheral blood in a concentration-dependent manner. The effects of LPA on adhesive properties are markedly attenuated in the presence of VPC-12249, an antagonist of LPA_1,2_ [14]. The pre-exposure of human pulmonary arterial endothelial cells to LPA also enhances the mRNA expression of adhesion molecules such as intracellular adhesion molecule 1 (ICAM-1), E-selectin, β_1_ integrin, and vascular cell adhesion molecule 1 (VCAM-1) [14] (Figure 3). LPA-induced expression of adhesion molecules in the endothelium may result in the recruitment of inflammatory cells in the respiratory system. Inhalation of LPA into guinea pigs augments the number of neutrophils and eosinophils in BALFs in a concentration-dependent manner, and inhalation of LPA enhances superoxide synthesis by these inflammatory cells compared with spontaneous production [17]. These effects of LPA on inflammatory cell recruitment and superoxide synthesis are attenuated in the presence of Y-27632, an inhibitor of Rho-kinase, in a concentration-dependent manner, indicating the involvement of the Rho/Rho-kinase pathway in these phenomena.

### 3.5. Clinical Relevance

#### 3.5.1. Pulmonary Hypertension

LPA is associated with various inflammatory diseases based on vascular dysfunction via the physiological action on endothelial and smooth muscle cells [44]. It is still unknown whether pulmonary smooth muscle cells are involved in LPA-induced pulmonary vascular remodeling. However, in a rat model of hypoxic pulmonary hypertension, LPA levels are markedly increased in serum and tissues, and fibroblast migration and the recruitment of mononuclear cells are facilitated in vessels in the respiratory system, resulting in pulmonary vascular remodeling [14] (Figure 3). Pulmonary vasculature remodeling probably causes hypertension in the pulmonary artery.

#### 3.5.2. Pulmonary Fibrosis

A clinical trial has demonstrated that LPA levels are markedly increased in patients with idiopathic pulmonary fibrosis, and that the suppression of LPA_1_ markedly attenuates the chemotactic effects on the fibroblasts in this fluid [15] (Figure 3). LPA is increased in BALFs and exhaled breath condensates in patients with idiopathic pulmonary fibrosis (IPF), and the expression of ATX with lysophospholipase D activity that converts LPC into LPA is increased in lung tissues from patients with IPF and fibrotic non-specific interstitial pneumonia [38] (Figure 3). The up-regulation of ATX is also closely associated with more progressive and irreversible forms of pulmonary fibrosis, such as IPF/usual interstitial pneumonia (UIP) and fibrosing nonspecific interstitial pneumonia (fNSIP) [38]. ATX-LPA is probably involved in the pathogenesis of pulmonary fibrosis. Moreover, the knockdown of LPA_2_ causes a reduction in bleomycin-induced lung injury and pulmonary fibrosis, and this phenomenon may be associated with a reduction in the LPA-induced expression of TGF-β and the activation and differentiation of fibroblasts [45]. Therefore, LPA_1,2_ is probably a novel therapeutic target for idiopathic pulmonary fibrosis, which responds to injury implicated in fibrosis.

#### 3.5.3. Asthma

When patients with asthma are challenged with an allergen, ATX/LPA levels are augmented in their BALFs, indicating that ATX/LTA may be involved in the allergic reaction related to asthma [9]. The inhibition of ATX causes a marked reduction in Th2 cytokines and allergic inflammation in the respiratory system in murine model of allergic asthma [46]. The antigen-challenged BALFs of ATX transgenic mice have a markedly increased number of inflammatory cells (mostly eosinophils) and Th2 cytokine levels (IL-4, IL-5) compared with wild-type mice; in contrast, mice in which the ATX-LPA pathway in blocked have a markedly decreased number of eosinophils and Th2 cytokine levels compared to wild-type mice [46]. Therefore, the ATX-LPA pathway may be a novel therapeutic target for asthma. The ATX-LPA pathway enhances superoxide levels [37], suggesting that LPA may be involved in the pathophysiology of asthma and chronic obstructive pulmonary disease (COPD), which are partly related to oxidative stress [47].

#### 3.5.4. Acute Respiratory Distress Syndrome: COVID-19

ATX levels in serum are increased in patients with severe COVID-19; increased serum levels of ATX are corelated with increased serum levels of IL-6, as well as endothelial dysfunction markers, such as soluble E-selectin, soluble P-selection, ICAM-1, and angiopoietin 2 [5] (Figure 3). Severe COVID-19 could lead to the development of ARDS with impaired gas exchange. ATX levels in serum and BALFs are associated with inflammatory and fibrotic biomarkers (IL-6, IL-8, TNF-a, fibronectin, etc.), and the severity of illness (the SOFA score and PaO_2_/FIO_2_ ratio) in patients with ARDS, indicating that ATX/LPA may be involved in the pathogenesis of ARDS [48]. The ATX-LPA pathway could be involved in the cytokine storm and subsequent pulmonary fibrosis in the pathophysiology of COVID-19. LPA-induced barrier dysfunction in endothelial cells may be involved in the pulmonary edema related to ARDS.

## 4. Lysophosphatidylcholine

### 4.1. Structure and Function

Lysophosphatidylcholine (LPC) is characterized by a glycerol backbone with the addition of a phosphate group, which is connected by choline, in the sn-3 position, and a fatty acid chain in either the sn-1 or sn-2 position, with variations being due to the different fatty acid chains. Choline attaches to a phosphate group in LPC; however, a hydroxyl group attaches to a phosphate group in LPA. Although extracellular LPC exhibits various physiological activities through binding to GPCRs, its specific receptors have not yet been identified as different from LPA and S1P. GPCRs, such as GPR40, GPR55, GPR119, and G2A, and Toll-like receptors have recently been shown to be specific receptors for LPC [4,49,50]. Another subset of GPCRs, such as GPR4 (G protein-coupledreceptor4), OGR1 (ovarian cancer G protein-coupled receptor 1), and TDAG8 (T cell death-associated gene 8), can also be listed as specific receptor of LPC [22,51]. LPC is synthesized by the action of PLA2 on phosphatidylcholine in the cell membrane, and it promotes inflammatory responses, including the induction of pro-inflammatory cytokine production, the increased expression of endothelial cell adhesion molecules and growth factors [52,53,54], the induction of monocyte chemotaxis [55], and macrophage activation [56], leading to the development of various diseases [4]. Since LPC is the main component of oxidized low-density lipoprotein (oxLDL), which accumulates in atherosclerotic plaques, the effects of LPC on endothelial cells and vascular smooth muscle cells in the arteries play an essential role in the progression of atherosclerosis [57,58]. LPC also increases the generation of reactive oxygen species (ROS), and it enhances oxidative stress [59], inducing not only atherosclerosis but also various inflammatory diseases, such as asthma and COPD [47]. Furthermore, LPC causes intracellular calcium dynamics by increasing external Ca^2+^ influx and Ca^2+^ release from the endoplasmic reticulum [60]. The effects of LPC occur through Ca^2+^ sensitization due to the Rho/Rho-kinase pathway [18,61]. Therefore, Ca^2+^ signaling is probably associated with LPC-induced biological responses and these pathways can be molecular targets for therapy in various inflammatory diseases [4,62].

### 4.2. Effects on Smooth Muscle

Previous reports have indicated that LPC causes endothelium-dependent relaxation in vascular smooth muscle through cGMP-related processes [63]. In contrast, the exposure of isolated aortic arteries to oxidized low-density lipoprotein markedly attenuate the relaxant action of acetylcholine on phenylephrine-induced contraction, indicating that lysolecithin probably impairs endothelium-dependent vasorelaxation in atherosclerosis [64] (Figure 4). LPC also inhibits endothelium-dependent relaxations and directly induces contractile responses in vascular smooth muscle [58] (Figure 4). Intracellular Ca^2+^ levels are increased by LPC in endothelium-denuded vascular smooth muscle. LPC-induced Ca^2+^ dynamics does not generate tension; however, it potentiates the contractile response [60]. Although oxLDL has no effects on vascular tone, LPC potentiates the angiotensin II-induced contraction, and this phenomenon is inhibited by Y-27632, a specific inhibitor of Rho-kinase [65] (Figure 4). The Rho/Rho-kinase pathway is involved in the intracellular signal transduction system related to LPC. In tracheal smooth muscle, LPC also does not generate tension with no change in intracellular Ca^2+^ levels, and pre-exposure to LPC does not potentiate methacholine-induced contraction. However, pre-exposure to LPC attenuates isoproterenol-induced relaxation against methacholine, with no change in intracellular Ca^2+^ levels. This β_2_-adrenegic desensitization is also recovered in the presence of Y-27632 [18]. These results indicate that LPC impairs the inhibitory effects of β_2_-adrenergic action on Ca^2+^ sensitization due to the Rho/Rho-kinase pathway [66,67].

Superoxide radicals oxidize low-density lipoprotein (LDL), and oxidized LDL has been shown to be an essential component in atherogenesis because LPC is accumulated during the oxidative modification of LDL. LPC enhances the proliferation of vascular smooth muscle cells through the activation of mitogen-activated protein kinase (MAPK)—extracellular signal-regulated kinase 1/2 (ERK1/2) processes, and this phenomenon is inhibited in the presence of protein kinase C (PKC) inhibitors (GF109203X, phorbol ester) and HMG-CoA reductase inhibitors (pitavastatin, atorvastatin, and fluvastatin) [68,69] (Figure 4). LPC-induced cell proliferation is caused by a release of fibroblast growth factor-2 from vascular smooth muscle cells [70]. Oxidized LDL and LPC with endothelin-1 synergistically enhance the proliferation of vascular smooth muscle cells; this phenomenon is inhibited in the presence of the antioxidant N-acetylcysteine, diphenylene iodonium (an inhibitor of NADPH oxidase), or PD098059 (an inhibitor of MAPK), indicating that LPC-induced cell proliferation is associated with the activation of redox-sensitive and MAPK—ERK1/2 pathways [71] (Figure 4). As an interaction between inflammatory cells and smooth muscle cells, treatment with LPC increases the level of heparin-binding epidermal growth factor-like growth factor (HB-EGF), a potent smooth muscle mitogen, in monocytes and macrophages, resulting in the initiation of the proliferation of vascular smooth muscle cells [72,73].

LPC facilitates the migration of human vascular smooth muscle cells at least in part through increased basic fibroblast growth factor production. This LPC-induced migration can be enhanced by platelet-derived growth factor-BB and endothelin-1, and can be attenuated by adrenomedullin and vitamin E [74]. Protein kinase C (PKC) is involved in the LPC-induced migration of vascular smooth muscle cells [75] (Figure 4). Since cell proliferation and cell migration are suppressed under the knockout conditions of LPA_1_ by siRNA and the inhibition of G_i_ proteins with pertussis toxin in human arterial smooth muscle cells [76], LPC-induced proliferation and migration is caused by the LPA_1_/G_i_/MAP kinase signaling pathway. LPC may lead to remodeling in the pulmonary artery through the proliferation and migration of smooth muscle cells through conversion into LPA by ATX. Therefore, LPC-ATX-LPA processes may be associated with the atherogenic action induced by ox-LDL.

### 4.3. Effects of Endothelial Cells

Since treatment with LPC increases the mRNA expression of the transient receptor potential channel (TRPC) in endothelial cells, LPC causes store-operated Ca^2+^ (SOCE) via the release Ca^2+^ from the endoplasmic reticulum (Ca^2+^ sparks), resulting in increases in intracellular Ca^2+^ levels [77] (Figure 4). LPC enhances the proliferation of endothelial cells with increases in the production of reactive oxygen species (oxidative stress) and the activation of large-conductance Ca^2+^-activated K^+^ (K_Ca_) channels [59] (Figure 4). LPC-induced K_Ca_ channel stimulation is probably associated with spontaneous transient outward currents (STOCs) due to Ca^2+^ sparks. This mechanism is different from that underlying β_2_-adrenergic action on K_Ca_ channels through G-protein/adenylyl cyclase/cAMP processes [78,79,80,81].

The oxLDL-based bioactive lipid (LPC) has effects on endothelial cells as a key regulator of adhesion marker expression. LPC can trigger the expression of intercellular adhesion molecule-1 (ICAM-1) and vascular cell adhesion molecule-1 (VCAM-1) in human umbilical vein endothelial cells via an orphan G protein receptor 2 (G2A)-related signaling pathway [54] (Figure 4). The CD11b/CD18 (Mac-1) molecule of the β_2_-integrin subfamily, as well as CD49d/CD29 (VLA-4) of the β_1_-integrin subfamily, have been reported to contribute to the adhesion of eosinophils to endothelial cells through binding to the endothelial ligands, ICAM-1 and VCAM-1 [82]. LPC also upregulates CD11b/CD18 to induce eosinophil adhesion through non-SOCE [83], potentially resulting in eosinophil infiltration into the airways in asthma [16] (Figure 4). The migration of eosinophils across the resting endothelium is caused by RANTES through the interaction of β_2_ integrins and ICAM-1; however, eosinophils express the β_1_ integrin, which mediates binding to VCAM-1. Moreover, β1 and X2 integrins are differentially regulated by chemoattractants in the adhesion of eosinophils [82]. It is generally considered that the transcription factor nuclear factor-kappaB (NF-κB) is involved in the expression of ICAM-1, VCAM-1, and E-selectin. However, the NF-κB inhibitor caffeic acid phenethyl ester (CAPE) and the antioxidant N-acetylcysteine partially inhibit LPC-induced adhesion molecules. GPR4, a class of GPCRs for LPC, causes the upregulation of adhesion molecules in endothelial cells through the cAMP/PKA/cAMP response element-binding protein pathway [84]. The adhesion of monocytes to endothelial cells is also caused by treatment with LPC through ICAM 1 transcription which is induced by mitochondrial reactive oxygen species and IL-33 can attenuate this endothelium activation by inhibiting LPC-induced mitochondrial reactive oxygen species [85].

LPC decreases transendothelial electrical resistance and increases endothelial permeability in pulmonary microvascular endothelial cells through cross-talk between PKC and Rho signals, indicating that LPC impairs the endothelial barrier function mediated by this interaction process [86] (Figure 4). GPR4 exists in the cell membrane of endothelial cells, and the siRNA-mediated silencing of GPR4 expression is associated with the prevention of the LPC-induced decrease in transendothelial resistance. Moreover, the knock down of GPR4 prevents both stress fiber formation and activation related to Rho in response to LPC. The results indicate that LPC-mediated endothelial barrier dysfunction is regulated by endogenous GPR4 in endothelial cells and that Rho-induced cytoskeleton reorganization is involved in this phenomenon [87] (Figure 4).

### 4.4. Clinical Relevance

#### 4.4.1. Atherosclerosis, Acute Respiratory Distress Syndrome

LPC stimulates superoxide production in endothelial cells through the NADH/NADPH oxidase-dependent mechanism [88]. It is generally considered that LPC is closely associated with cardiovascular diseases due to atherosclerosis because superoxide anions inactivate anti-atherogenic molecules. LPC is the major component of modified LDL and oxidized LDL, which play significant roles in the development of atherosclerotic plaques via monocyte adhesion and smooth muscle proliferation [52,53] (Figure 4). LPC enhances vascular permeability through endothelial dysfunction due to cytoskeleton reorganization [87] (Figure 4), potentially resulting in various inflammatory respiratory diseases, such as pulmonary edema due to ARDS.

#### 4.4.2. Asthma

LPC may also be involved in the pathophysiology of asthma through the recruitment of eosinophils to the respiratory system. LPC levels are increased in the BALFs of allergen-challenged mice [89] (Figure 4). Plasma LPC levels markedly rise in patients with asthma and rhinitis compared with normal subjects, and this phenomenon is closely correlated with increased airway responsiveness to histamine in most patients with asthma [90] (Figure 4). Exposure to LPC induces eosinophil infiltration into the lungs, and it enhances respiratory resistance in guinea pigs [16] (Figure 4). The recruitment of eosinophils to the airways is an essential feature for chronic allergic inflammation such as asthma. LPC is synthesized by PLA2 with free fatty acids such as arachidonic acid, a substrate for the synthesis of biologically active eicosanoids (prostaglandins and leukotrienes). The activation of PLA2 is closely implicated in the pathophysiology of asthma. Furthermore, LPC may contribute to the pathophysiology of COPD because LPC is associated with emphysema through the dysfunction of pulmonary alveolar and the epithelium due to the enhancement of permeability and apoptosis [91,92]. LPC-induced oxidative stress may also result in the development of COPD [47].
Figure 4The effects of lysophosphatidylcholine on the constituent cells and related molecular mechanisms in pulmonary blood vessels. Disease arising from pathophysiology based on physiological activities induced by lysophosphatidic acid. SOCE: store-operated calcium entry; PKC: protein kinase C; MAPK: mitogen-activated protein kinase; ERK: extracellular signal-regulated kinase; PI3K: phosphoinositol 3-kinase; ICAM-1: intercellular adhesion molecule 1; VCAM-1: vascular cell adhesion molecule-1. The 1st row: the constituent cells; the 2nd row: molecular mechanisms; the 3rd row: biological activities; the 4th row: pathophysiology; the 5th row: diseases related to pulmonary vasculatures. Illustrated based on Refs. [4,16,18,52,53,54,57,58,59,60,61,62,64,65,66,67,68,69,70,71,75,76,77,78,79,80,81,82,83,86,87,88,89,90].
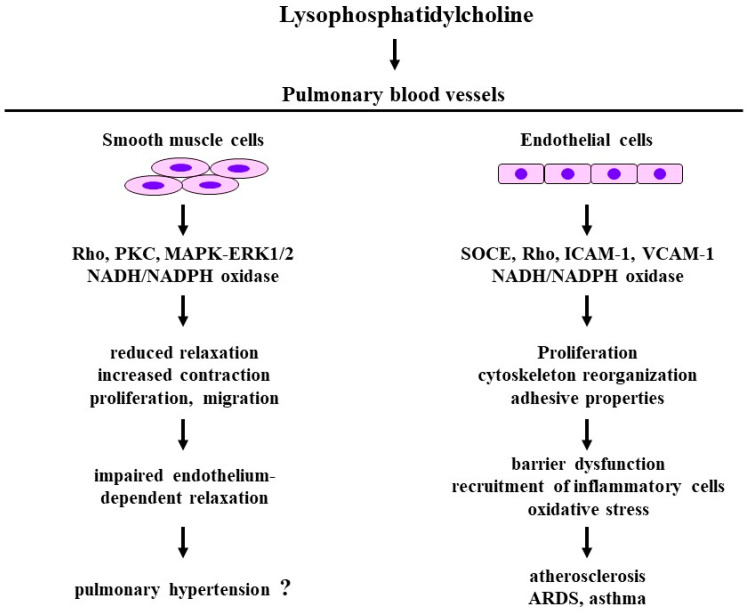



#### 4.4.3. Pulmonary Hypertension

Since LPC causes the contraction and proliferation of smooth muscle cells, it may be related to the development of pulmonary hypertension due to the stenosis of pulmonary arteries. However, the roles of LPC in the pathogenesis of these respiratory diseases are still to be determined.

## 5. Sphingosine 1-Phosphate

### 5.1. Structure and Function

Sphingosine 1-phosphate (S1P), which is a bioactive metabolite of sphingolipids, is rapidly generated from sphingosine via the action of sphingosine kinase (SphK) in the plasma membrane, and it is a type of lysophospholipid with a sphingosine backbone. S1P consists of different carbon long-chain base lengths; however, the most common structure is d18:1 (double hydroxy groups, 18-carbon chain length and 1 unsaturated bond), which accounts for approximately 80% of the total S1P in plasma. S1P is transported with high-density lipoprotein cholesterol in about 60% of plasma [93], mainly through the promoter superfamily transporter 2b (Mfsd2b) [94].

Sphingolipids, originating from sphingomyelin in the plasma membrane, are metabolized, and sphingosine is synthesized from ceramides via the action of ceramidase. SphK consists of two isozymes, namely, sphingosine kinase 1 (SphK1) and 2 (SphK2). S1P is generated on the intracellular side, and it is transported to the extracellular side by a multipass transmembrane protein, spinster homolog 2 (Spns2) [95]. Extracellular S1P acts on the endothelial and smooth muscle cells in vessels as lipid mediators through binding to their specific receptors (GPCRs), which are classified into five subtypes, namely, S1P_1–5_, leading to various biological activities, such as proliferation, migration, cytoskeletal reorganization, and differentiation in smooth muscle and endothelial cells [96]. Hence, S1P is closely associated with the regulation of angiogenesis and vascular function [97,98]. S1P_1_, which is expressed in endothelial cells, is connected to the inhibitory G protein of adenylyl cyclase (G_i_), resulting in the activation of Ras/ERK processes and PI3K/Ark processes. S1P_1,2_, which are expressed in endothelial and smooth muscle cells, can be connected to various G proteins, for example, not only G_i_ but also G_12/13_ and G_q_, resulting in the activation of Rho/Rho-kinase processes and PKC/Ca^2+^ dynamics, respectively.

S1P is released from a variety of cells, for example, not only hematopoietic cells (red blood cells and platelets) but also endothelial cells, in a constitutive manner [99]. Therefore, a large amount of S1P is released from these cells, resulting in a high concentration of S1P in plasma [100], and the concentration of S1P in plasma is increased to ~1 μmol/L in healthy subjects [99,101]; in contrast, the concentration of S1P in interstitial fluid is generally ~100 nmol/L [101]. A large concentration gradient in S1P is maintained between vascular and extravascular compartments, probably resulting in the physiological activities induced by extracellular S1P [100]. Therefore, external S1P acts on constituent cells in the airways, and it exerts various physiological activities, resulting in the development of pulmonary diseases such as pulmonary hypertension [102].

### 5.2. Effects on Endothelial Cells

A previous report has demonstrated that S1P causes a concentration-dependent increase in transmonolayer electrical resistance across endothelial cells and that this effects of S1P is reduced after treatment with EDG_1_ antisense, pertussis toxin (an inhibitor of G_i_), Y-27632 (an inhibitor of Rho-kinase), and cytochalasin B (an inhibitor of actin polymerization) [97] (Figure 5). External S1P acts on endothelial cells and improves barrier integrity, and the endothelial barrier function is caused by S1P_1_/G_i_ processes and the Rac-induced recruitment of actin filaments [97] (Figure 5). In a previous study, in pS1Pless mice generated by both SphK1 and SphK2 knockout, the plasma concentration of S1P was markedly reduced (almost undetectable), and vascular leak in response to serotonin and histamine was markedly augmented compared with wild-type mice [103]. Moreover, although all wild-type mice survived after passive systemic anaphylaxis, 44% of pS1Pless mice died with increased lung weight and extracellular fluid within 30 min of the administration of these anaphylaxis-induced agents [103].

External S1P also acts on human umbilical vein endothelial cells (HUVECs) and it enhances expression of adhesion molecules such as, VCAM-1 andICAM-1, mainly through S1P_3_ [104] (Figure 5). This effect of S1P is caused by nuclear factor kappa-B activation, and it is abolished by treatment with pertussis toxin, indicating the involvement of the S1P/Gi processes in this physiological activity. S1P causes an increase in the intracellular concentration of Ca^2+^ through non-selective cation channels in HUVECs, and pre-exposure to pertussis toxin markedly inhibits this S1P-induced Ca^2+^ entry [105] (Figure 5). In contrast, S1P significantly enhances the expression of VCAM-1 and ICAM-1 in human pulmonary microvascular endothelial cells (HPMVECs), but only slightly enhances the expression of ICAM-1 [13] (Figure 5). Furthermore, in the analysis of an adhesion assay, pre-exposure to S1P was found to cause concentration-dependent increases in the adhesion of human eosinophilic leukemic cell line cells to HPMVECs [13]. These effects of S1P on HPMVECs are inhibited in the presence of Y-27632 and pre-incubation with pertussis toxin, indicating that Ca^2+^ sensitization related to the G_i_/Rho-kinase pathways may be involved in this physiological activity, perhaps S1P_2_. Ca^2+^ dynamics are also affected in the presence of S1P in HPMVECs. S1P causes a phasic increase in the intracellular concentration of Ca^2+^ through stretch-induced Ca^2+^ entry [106]. Both Ca^2+^ sensitization and Ca^2+^ dynamics are associated with the effects of S1P on endothelial cells.

### 5.3. Effects of Smooth Muscle

S1P may cause the relaxation of vascular smooth muscle since endothelial cell nitric oxide synthase (eNOS) is activated in the endothelium through the PI3K/Akt process, which occurs downstream of the S1P_1–3_/G_i_ signal pathways. However, since the signal transduction pathway of S1P (S1P_2_ and S1P_3_) is involved in PLC/IP_3_ and Rho/Rho-kinase, which are related to Ca^2+^ dynamics and Ca^2+^ sensitization, respectively, S1P can generate tension in vascular smooth muscle with Ca^2+^ signal pathways similar to trachea [12,107]. Previous reports have demonstrated that S1P causes contraction in coronary artery with an increase in intracellular concentration of Ca^2+^ (Ca^2+^ dynamics) and Rho-kinase activation (Ca^2+^ sensitization) [108] (Figure 5), and that S1P also causes contraction in the aorta with an increase in the intracellular Ca^2+^ concentration through SOCE [109] (Figure 5). The Ca^2+^ dynamics due to S1P/SOCE processes are associated not only with constriction but also with cell proliferation in vascular smooth muscle, leading to the dysfunction of vessels. The SphK1/S1P process is also associated with cell proliferation related to growth factors (TGF-β_1_ and PDGF) in pulmonary arterial smooth muscle [110,111] (Figure 5). Furthermore, S1P facilitates not only proliferation but also migration in pulmonary arterial smooth muscle cells through Ca^2+^ dynamics due to TRPC [112] and Ca^2+^ sensitization due to RhoA/Rho-kinase processes [113,114] (Figure 5). Moreover, PI3K/Akt and phospholipase C, which are downstream of G_i_ and Gq, respectively, are involved in S1P-induced cell migration in pulmonary arterial smooth muscle [115,116] (Figure 5). Therefore, external S1P is closely related to alterations in the structure and function of pulmonary vessels [102].

### 5.4. Effects on Fibroblasts

S1P does not have effects on fibroblast migration, but it augments fibronectin-induced chemotaxis in HFL-1 cells (a lung fibroblast cell line) through S1P_2_/Rho-kinase processes and focal adhesion kinase (FAK) phosphorylation [117]. This S1P-induced fibroblast chemotaxis is not associated with S1P_1_ or S1P_3_. FTY720-P, a non-selective agonist of S1P receptors, enhances PI3K/Akt and ERK1/2 processes in human lung fibroblasts through the activation of S1P_2_ and S1P_3_, leading to the stimulation of extracellular matrix (ECM) synthesis [118] (Figure 5). The administration of bleomycin and TGF-β causes increases in SphK1 and S1P levels in embryonic lung fibroblasts [119], and S1P cross-talks with TGF-β to amplify the loop between S1P and TGF-β with other cytokines to facilitate fibrogenesis [120,121]. The SphK1/S1P pathway activates lung fibroblasts through the stimulation of Yes-associated proteins (YAPs) related to TGF-β, the generation of mitochondrial reactive oxygen species (mtROS) related to bleomycin, and the expression of fibronectin (FN) and alpha-smooth muscle actin (α–SMA) [122].

### 5.5. Clinical Relevance

#### 5.5.1. Pulmonary Hypertension

S1P levels in serum are increased in patients with pulmonary hypertension, and the expression levels of both SphK1 and SphK2 in remodeled pulmonary arteries are also increased [123]. The SphK1/S1P processes trigger hypoxia-induced vasoconstriction in human pulmonary arterial smooth muscle cells through Ca^2+^ dynamics due to TRPC and Ca^2+^ sensitization caused by Rho-kinase, probably resulting in pulmonary hypertension [124] (Figure 5). This disease is characterized by markedly elevated pulmonary arterial resistance, which is due to the sustained constriction and excessive remodeling of pulmonary artery. Since S1P causes the contraction and proliferation of pulmonary arterial smooth muscle cells, S1P is probably involved in the development of this disease [125]. SphK1 and S1P levels are markedly augmented in the lungs and pulmonary arterial smooth muscle cells of patients with pulmonary hypertension compared with healthy subjects, whereas SphK2 levels are not [125] (Figure 5). SphK1-deficient mice are also less likely to develop pulmonary arterial hypertension than wild-type mice, and pulmonary arterial smooth muscle cells of SphK1-deficient mice are less likely to proliferate than those of wild-type mice [126]. However, these phenomena are not mimicked in SphK2-deficient mice. Moreover, verteporfin, an inhibitor of YAP1, reduces the S1P-induced proliferation of pulmonary arterial smooth muscle cells and hypoxia-induced pulmonary arterial hypertension [126], indicating that YAP1 is associated with the development of SphK1-induced pulmonary hypertension. Therefore, the inhibition of SphK1 and YAP1 may be a therapeutic target for this disease.

#### 5.5.2. Acute Respiratory Distress Syndrome

ARDS, which is caused by pneumonia, sepsis, trauma, etc. [127], is essentially characterized by endothelial barrier dysfunction (an increase in vascular leak). S1P probably has protective effects on this pathophysiology of ARDS [101] because the intravenous administration of S1P results in a marked reduction in endothelial barrier dysfunction in animal models of endotoxin-induced acute lung injury (mild ARDS) [128] (Figure 5). Lipopolysaccharide (LPS)-induced vascular leak is more potent in SphK1 knockout mice than in wild-type mice [129], and an intratracheal or intravenous administration of S1P or SEW-2871, an agonist of S1P_1_, reduces endothelial barrier dysfunction in murine models of LPS-induced ARDS, whereas SB-649146, an antagonist of S1P_1_, reverses this effect of SEW-2871 [130]. S1P is also effective for vascular leak through endothelial barrier dysfunction related to anaphylaxis, and delayed histamine clearance is observed in SphK1 knockout mice [131] (Figure 5).

#### 5.5.3. Asthma

Previous reports have demonstrated that S1P is probably associated with asthma [132,133] (Figure 5). When patients with asthma are exposed to an allergen, S1P levels increase in the BALFs of these patients [10] (Figure 5). Regarding the mechanism, it is generally considered that the antigen/IgE/FcεRI pathway in mast cells causes an increase in S1P levels through SphK1 activation [134,135]. S1P acts on eosinophils to facilitate chemotaxis and recruitment through its receptors and CCR3 [136] and S1P also acts on endothelial cells to enhance the expression of adhesion molecules, resulting in recruitment to the lungs via the adhesion of eosinophils to endothelial cells [13] (Figure 5). These results indicate that S1P is involved in allergic reactions and airway eosinophilic inflammation in asthma. Moreover, S1P causes contraction and an increase in the response to muscarinic agonists in airway smooth muscle [12,137], indicating that S1P is involved in airflow limitation and airway hyperresponsiveness in asthma. The administration of SK1-I, a specific inhibitor of SphK1, reduces allergen-challenge-induced eosinophilic inflammation and hyperresponsiveness in the airways through the suppression of NF-kB in a mast-cell-dependent murine model of allergic asthma [138] (Figure 5).

#### 5.5.4. Pulmonary Fibrosis

Since the SphK1/S1P pathway stimulates lung fibroblasts through activated YAP1 and synthesized mtROS related to TGF-β, as described in Section 5.4, these processes are essential mechanisms for the development of pulmonary fibrosis. Clinical trials have demonstrated that S1P levels in BALFs are markedly augmented in patients with IPF compared to healthy subjects [139,140], and that S1P levels in serum are also markedly augmented in patients with IPF compared to healthy subjects [139] (Figure 5).
Figure 5The effects of sphingosine 1-phosphate on the constituent cells and related molecular mechanisms in pulmonary blood vessels. Disease arising from pathophysiology based on physiological activities induced by lysophosphatidic acid. SOCE: store-operated calcium entry; TRPC: transient receptor potential channels; NSCC: non-selective cation channels; ERK: extracellular signal-regulated kinase; PI3K: phosphoinositol 3-kinase; PLC: phospholipase C; ICAM-1: intercellular adhesion molecule 1; VCAM-1: vascular cell adhesion molecule-1. The 1st row: the constituent cells; the 2nd row: molecular mechanisms; the 3rd row: biological activities; the 4th row: pathophysiology; the 5th row: diseases related to pulmonary vasculatures. Illustrated based on Refs. [10,13,96,97,101,102,104,105,106,107,108,109,110,111,112,113,114,115,116,117,118,122,123,124,125,127,128,129,130,131,132,133,134,135,136,137,138,139].
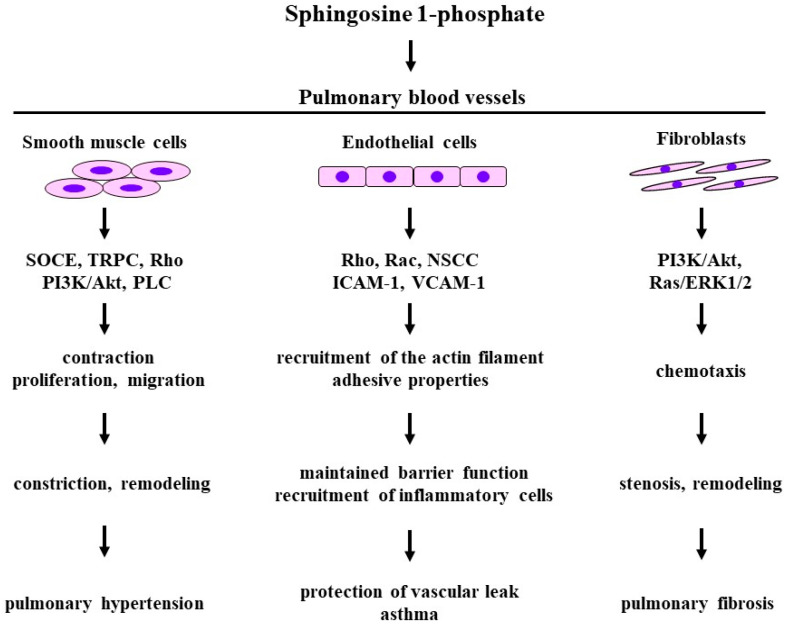



## 6. Conclusions

Extracellular lysophospholipids (LPA, LPC, and S1P) act as lipid mediators, and they have various effects (contraction, proliferation, migration, and cytoskeleton reorganization) on the constituent cells in the respiratory system, such as endothelial cells, smooth muscle cells, and fibroblasts. These physiological activities are probably involved in the pathophysiology of inflammatory respiratory diseases, such as pulmonary edema due to ARDS and anaphylaxis; pulmonary hypertension; pulmonary fibrosis; and asthma through endothelial barrier dysfunction, pulmonary vasculature remodeling, and the recruitment of eosinophils. However, the clinical relevance of lysophospholipids in inflammatory diseases related to pulmonary vessel dysfunction is still unknown. Development of precision medicine focused on lysophopholipids as treatable traits is probably needed to advance the management and therapy of these inflammatory pulmonary diseases in the near future. 

## Figures and Tables

**Figure 1 biomedicines-12-00124-f001:**
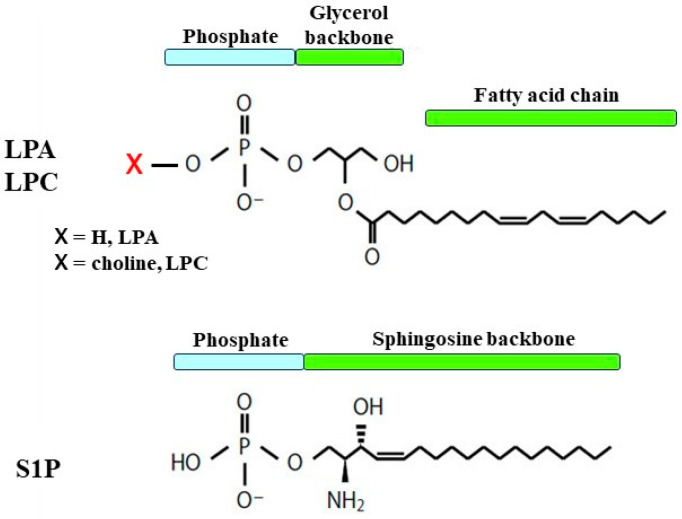
The representative structures of lysophospholipids. LPA: lysophosphatidic acid; LPC: lysophosphatidylcholine; S1P: sphingosine 1-phosphate. The number of carbons and double bonds varies depending on each fatty acid.

**Figure 3 biomedicines-12-00124-f003:**
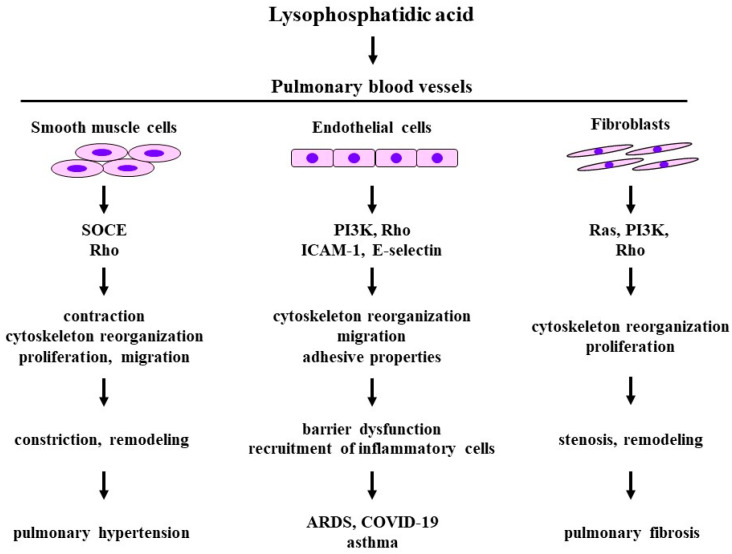
The effects of lysophosphatidic acid on the constituent cells and related molecular mechanisms in pulmonary blood vessels. Disease arising from pathophysiology based on physiological activities induced by lysophosphatidic acid. SOCE: store-operated calcium entry; PI3K: phosphoinositol 3-kinase; ICAM-1: intercellular adhesion molecule 1. The 1st row: the constituent cells; the 2nd row: molecular mechanisms; the 3rd row: biological activities; the 4th row: pathophysiology; the 5th row: diseases related to pulmonary vasculature. Illustrated based on Refs. [5,9,11,14,15,17,24,29,30,32,33,34,35,36,37,38,39,40,41,42,43,44,45,46,47,48].

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
