# Peer review of "Involvement of Lysophospholipids in Pulmonary Vascular Functions and Diseases"

_biomedicines, 2024, doi:10.3390/biomedicines12010124_

Round 1
Reviewer 1 Report
Comments and Suggestions for Authors
To the authors:
1. General comments:
The review entitled “Involvement of lysophospholipids in pulmonary vascular functions and diseases” shows the roles of 3 important lipids; lysophosphatidic acid, lysophosphatidylcholine, and sphingosine 1-phosphate. From each of them, structure and function, effects on different cell types (smooth muscle, fibroblast, endothelial cells), and clinical relevance are described. The manuscript is well-written, but I found some major and minor mistakes that authors should be corrected before publication.
2. Specific comments for revision: b) major
1. Currently, the main classification of lipids, is the one proposed by LIPID MAPS (doi: 10.1016/j.bbalip.2011.06.009), which states that S1P is a sphingolipid rather than a lysophospholipid. So, I would suggest encompassing these lipids as membrane's lipids rather than lysophospholipids.
2. Please review the aim of the review and try to condensate the information, in my opinion is quite large, and there are several lines that are difficult to follow (please see below the comments).
Minor comments:
3. Line 15. Which lipid do you refer to as the adhesion molecule?
4. Line 20. Define ARDS
5. Line 59. Define BALF, and later only use this abbreviation when needed.
6. Line 95. Please check the spelling of phosphorylation
7. Line 108, 109, 152, 157, etc. Please unify the text, I would suggest not using subindex for LPA1,2,3.
8. Line 110. Same for S1P1-5.
9. Line 543 and 379. Please unify when to use the subindex for Gi
10. Lines 188-191. The sentence is too large, and the meaning is not understandable. Please check and amend.
11. Line 194. Add a reference.
12. Line 269. Define IPF/UIP
13. Line 286. Define COPD
14. Line 293. Use the abbreviation ARDS as was already defined
15. Line 293 -296. The sentence is too large, and the meaning is not understandable. Please check and amend.
16. Lines 404-406. Please review the sentence, is difficult to understand.
17. Line 444. Use the abbreviation of BLAF only
18. Line 450. Look at the word sech
19. Lines 505-508. The sentence is too large, and the meaning is not understandable. Please check and amend.
20. Line 578. Define PAH, Pas, HPH.
21. Line 588. Please specify which disease?
22. Line 599. Use the abbreviation ARDS
23. Line 599-602. Please check the sentence, the meaning is repeated.
24. Line 603. Please define ALI
25. Line 615. Use the abbreviation for BALF
26. Line 632,633. Use the abbreviation for IFP
Comments on the Quality of English Language
There are several lines that are difficult to follow (please see my comments).
Author Response
Response to Reviewer #1
Major comments
- I understand well what the reviewer says. However, even recently, several reports also refer S1P as a lysophospholipid. The main articles are shown below. I guess that “membrane’s lipids” may not be so attractive and difficult to understand for readers. Therefore, it is appropriate to leave the title unchanged. What do you think?
The sentence shown below was added to at the beginning of “INTRODUCTION”.
“Lysophospholipids, which are phospholipids that have one acyl group, are classified into two types, i.e., those with a glycerol backbone and a sphingosine backbone.”
Kano K, Aoki J, Hla T.
Lysophospholipid Mediators in Health and Disease.
Annu Rev Pathol. 2022;17: 459-483. doi: 10.1146/annurev-pathol-050420-025929.
Abdel-Latif A et al.
Lysophospholipids in coronary artery and chronic ischemic heart disease.
Curr Opin Lipidol. 2015; 26: 432-7. doi: 10.1097/MOL.0000000000000226
Nussbaum C et al.
Sphingosine-1-phosphate receptor 3 promotes leukocyte rolling by mobilizing endothelial P-selectin.
Nat Commun. 2015; 6: 6416. doi: 10.1038/ncomms7416.
Proia RL, Hla T.
Emerging biology of sphingosine-1-phosphate: its role in pathogenesis and therapy.
J Clin Invest. 2015; 125: 1379-87. doi: 10.1172/JCI76369.
Choi JW, Chun J.
Lysophospholipids and their receptors in the central nervous system.
Biochim Biophys Acta. 2013; 1831: 20-32. doi: 10.1016/j.bbalip.2012.07.015.
- There are currently no effective treatments for chronic inflammatory diseases related to pulmonary vessels such as pulmonary hypertension, pulmonary fibrosis, ADRS, and asthma.
In this review, we state not only effects of lysophospholipids on the pulmonary vessels but also involvement of lysophospholipids in the pathophysiology of these diseases to establish a novel therapy for these incurable respiratory diseases.
In this revised manuscript, the text is fixed according to an English speaker.
Minor comments
- LPA, LPC, and S1P enhances expression of adhesion molecules in the pulmonary endothelial cells.
- Changed to “acute respiratory distress syndrome (ARDS)”. Acute lung injure is not used.
- Changed to “bronchoalveolar lavage fluids (BALFs)”.
- Changed to “phosphorylation”.
- Changed to LPA1-3.
- Changed to S1P1-5.
- Changed to Gi.
- “When proliferation and migration are investigated using an electrical cell-substrate im-pedance sensor (ECIS), an inhibitor of ATX (PF8380) and an inhibitor of LPA receptors (Ki16425) attenuate proliferation and migration in carotid arterial smooth muscle cells in mice [38]”.
- “Illustrated based on refs [5], [9], [14], [15], [17], [24], [29-30], [32-49]” was added to the legend in Figure 3. “Illustrated based on refs [4], [16], [18], [54-56], [59-64], [66-73], [77-87], [90-94]” was added to the legend in Figure 4. “Illustrated based on refs [10], [13], [100], [101], [105], [106], [108-122], [126’-129], [131-143]” was added to the legend in Figure 5.
- Change to IPF/usual interstitial pneumonia (UIP) and fibrosing nonspecific interstitial pneumonia (fNSIP). “idiopathic pulmonary fibrosis (IPF)” is shown on the line above it in the same section.
- Changed to “chronic obstructive pulmonary disease (COPD)”.
- “acute respiratory distress syndrome (ARDS)” was changed to ARDS.
- “ATX levels are associated with inflammatory and fibrotic mediators such as IL-6, IL-8, TNF-a, fibronectin etc., in patients with ARDS, indicating that ATX/LPA may be involved in fibrotic component of ARDS” was changed to “ATX levels in serum and BALFs are associated with inflammatory and fibrotic biomarkers (IL-6, IL-8, TNF-a, fibronectin etc.), and the severity of illness (the SOFA score and PaO2/FIO2 ratio) in patients with ARDS, indicating that ATX/LPA may be involved in the pathogenesis of ARDS [49]”.
- Changed to “The CD11b/CD18 (Mac-1) molecule of the b2-integrin subfamily, as well as CD49d /CD29 (VLA-4) of the b1-integrin subfamily, have been reported to contribute to the adhesion of eosinophils to endothelial cells through binding to the endothelial ligands, ICAM-1 and VCAM-1 [86]”.
- Changed to “BALFs”.
- Changed to “such”.
- Changed to “In a previous study, in pS1Pless mice generated by both SphK1 and SphK2 knockout, the plasma concentration of S1P was markedly reduced (almost undetectable), and vascular leak in response to serotonin and histamine was markedly augmented compared with wild-type mice [107].”.
- Changed to “S1P levels in serum are increased in patients with pulmonary hypertension, and the expression levels of both SphK1 and SphK2 in remodeled pulmonary arteries are also in-creased [127].”
- Changed to “pulmonary hypertension”.
- Changed to “ARDS”.
- Changed to “ARDS, which is caused by pneumonia, sepsis, trauma, etc. [131], is essentially char-acterized by endothelial barrier dysfunction (an increase in vascular leak). S1P probably has protective effects on this pathophysiology of ARDS [105], because the intravenous administration of S1P results in a marked reduction in endothelial barrier dysfunction in animal models of endotoxin-induced acute lung injury (mild ARDS) [132] (Figure 5).”.
- Changed to “acute lung injury (mild ARDS)”.
ALI is most recently reclassified as moderate or mild acute respiratory distress syndrome (ARDS)
Mowery NT, Terzian WTH, Nelson AC.
Acute lung injury.Curr Probl Surg. 2020 May;57(5):100777.
- Changed to “BALFs”.
- Changed to “IPF”.
Reviewer 2 Report
Comments and Suggestions for Authors
The overall structure of the article needs to be reorganized to make it easier for readers to understand.

It's better to avoid long sentences that will confuse the readers.
Author Response
Response to the reviewer’s comments
The text has been fixed to avoid long sentences in this revised manuscript, according to the reviewer’s suggestion. Please take a look at it.

Reviewer 3 Report
Comments and Suggestions for Authors
This review was addressed to evidence the molecular mechanisms by which extracellular lysophospholipids (ELPLs) such as lysophosphatidic acid (LA), lysophosphatidylcholine (LPC) and sphingosine 1-phosphate (SGP) are involved, as lipid mediators, in a series of effects at the level of constituent cells in the respiratory tract. These cells include smooth myocytes, endothelial cells and fibroblasts where ELPLs appear to induce alterations that may play a role for the development of pathologies like ARDS, pulmonary hypertension and fibrosis, and asthma.
Introduction deals with pharmacological considerations about ELPLs and their G-coupled receptors allowing to influence various cell functions (e.g. migration, apoptosis and proliferation). Other pathophysiologic processes, like inflammation and oxidative stress, are conditioned by ELPLs so that the ELPLs-activated cell pathways are likely to represent an opportunity for innovative drug interventions mainly in case of respiratory diseases. Authors then range over structure and physiology of ELPLs with particular emphasis on the transductional pathways activated by these molecules. Subsequently, LA, LPC and SGP are detailed as well as their structure and function, and effects on smooth myocytes, fibroblasts and endothelial cells are concerned with adequate consideration of the related clinical pictures like pulmonary hypetension and fibrosis, asthma, ARDS, atherosclerosis and COVID-19 infection. It was concluded that a thorough knowledge of the ELPLs-induced cell effects, following activation of selected molecular pathways, may represent an adequate pathogenetic mainstay for the development of precision drugs able to efficaciously counteract the above pathologies.
Overall, this review has been well organized and well prepared, thus offering an appreciable detailed synthesis about chemical, pharmacological, physiological and pathological aspects concerning ELPLs. The manuscript is enriched with five figures provided with representative efficacy and executive accuracy. The reported 143 references are pertinent and exhaustive. Lexicon, sentence fluency and “English” style are adequate. Some minor editorial remarks: For instance: -line 18:…causes…=…cause…; line 21:…are cause…=…cause…; lines 69-70: …is associated with the allergic rection due to the allergen/IgE/mast cells processes…=…may induce a type I (IgE-dependent) allergic response…; lines 130-132:…lead to the…activities…=…lead to…effects such as…; lines 183-185:…it is still unknown…that LPA directly generates…is associated with…=…it is debatable whether LPA directly-induced contraction…is involved in the…; line 203:…the cytoskeletal…=…cytoskeletal…; lines 205-206:…low concentration (10%) oxygen inhalation for 3 weeks causes in rats…; an so on so that the honourable Authors should “leak out” the text in order to avoid several phraseological inaccuracies, thus making sentences more nimble and meaningful.
Comments on the Quality of English Language
Quality of English language may be thought to be adequate with the exception of several editorial refinements already suggested to the Authors.
Author Response
Response to Reviewer #3
Minor comments
line 18: “causes” was changed to “induce”
line 21: “are cause” was changed to “induce”
lines 69-70: “is associated with the allergic rection due to the allergen/IgE/mast cells processes.” was changed to “may induce a type I (IgE-dependent) allergic response.”
lines 130-132: ”activities” was changed to ”effects”
lines 183-185: Changed to “it is debatable whether LPA’s directly-induced contraction in pulmonary vascular smooth muscle is involved in the pathophysiology of pulmonary hypertension.”.
line 203: “the cytoskeletal” was changed to “cytoskeletal”
lines 205-206: Changed to “Both the serum levels of LPA and the perivascular expression of LPA are increased in a rat model of hypoxic pulmonary hypertension, which is characterized by structural changes in the vascular wall of pulmonary arteries [14]."
Round 2
Reviewer 1 Report
Comments and Suggestions for Authors
To the authors:
Dear authors, I would like to thank you for taking into account the suggestions and comments. I consider that the manuscript has improved substantially. I only have a couple of minor comments, but after that, I recommend it for publication.
Minor comments:
Line 99-100. Please check the term “long-chain fatty acids”, is it possible that intermediate C8:0-C12:0 can be formed?
Line 125. Check the sequence in the numbers for Edg.

The english has been greatly improved.
Author Response
Response to the reviewer 1
I appreciate your excellent review and important comments.
Minor comments
Line 99-100.
Changed to “a fatty acid chain” according to the reviewer’s suggestion.
“long” was deleted.
Line 125.
I checked the sequence in the numbers for Edg carefully. It is right.
Reviewer 2 Report
Comments and Suggestions for Authors
All the issue has been corrected.
Author Response
Response to the reviewer 2
I appreciate your excellent review and important comments.